# The impact of 8-week re-training following a 14-week period of training cessation on Greco-Roman Wrestlers

**Burhan Demirkıran, Ali Işın, Yılmaz Sungur, Tuba Melekoğlu** *

Department of Coaching Education, Faculty of Sport Sciences, Akdeniz University, Antalya, Turkey

* tmelekoglu@akdeniz.edu.tr

## Abstract

### Background

This study aimed to examine the changes in physical and physiological conditions in elite wrestlers from the Turkish National Wrestling Team, who experienced 14 weeks of restricted physical activity during the COVID-19 lockdown, followed by an 8-week period of retraining and competition.

### Methods

Twenty male elite wrestlers from the National Greco-Roman Wrestling Team participated in the research. Heart Rate Variability values were measured during the training cessation period and for 8 weeks of subsequent training and then interpreted for training periods with different workloads. Body fat percentage values, initially measured during training cessation, were recorded at 2-week intervals during the training period. To determine the fitness status of wrestlers, the Specific Wrestling Fitness Test was used before and following the 8 weeks of training period.

### Results

A gradual decrease in both body fat percentage and weight was observed throughout the course of the training period. The SWFT scores showed significant improvements ($31.40 \pm 2.91$ vs. $37.40 \pm 3.22$) following the training period. Heart rate variability decreased during the detraining period, progressively improved throughout the 8-week retraining, and subsequently declined during the competition phase, reaching levels similar to those observed during training cessation.

### Conclusions

Our results suggest that athletes undergo identical reactions in their autonomous nervous system during both competition and training cessation period. Obtaining a comprehensive understanding of these changes can enable coaches and athletes

**Data availability statement:** All dataset files are available from the Figshare database: https://doi.org/10.6084/m9.figshare.29047097.

**Funding:** Initials of author: TM Project Number: TYL-2021-5677 Funder:Scientific Research Projects Coordination Unit of Akdeniz University. Website: https://bap.akdeniz.edu.tr/tr Sponsors or funders didn't play any role in the study design, data collection and analysis, decision to publish, or preparation of the manuscript.

**Competing interests:** The authors have declared that no competing interests exist.

to make accurate decisions in order to optimize training adaptations and attain overall athletic success. Furthermore, over a period of eight weeks following a long non-training period, significant improvements in athletes' body fat, muscle mass and wrestling performance can be achieved along with training. Moreover, HRV monitoring revealed that autonomic nervous system balance was compromised during both the 14-week detraining and the subsequent competition phases, underlining the need for careful training load management to optimize recovery and performance readiness.

## Introduction

Elite athletes engage in continuous, high-intensity training that is planned and systematic. This approach is essential for achieving success in international competitions, sustaining achievements, and maintaining peak performance over extended periods. As a result of these high-intensity training regimens, the physiological adaptations observed in elite athletes differ significantly from those seen in sedentary individuals and less-trained athletes. Therefore, athletes who aim to perform at the highest level not only seek to further develop these adaptations but also strive to maintain them. To preserve or enhance these physiological adaptations, it is essential to apply training stimuli at a consistent and adequate level. While excessive training stimuli may lead to overtraining, insufficient or completely discontinued training can result in performance declines known as detraining. [1,2].

High-level competitive athletes who wish to avoid performance loss typically minimize the duration of detraining by limiting breaks from training. However, in certain circumstances—such as injury, chronic conditions, or pregnancy in female athletes—athletes may be forced to stop training. During periods of partial or complete training interruption, the body's physiological responsiveness to exercise stimuli may become diminished or delayed, leading to a slower return to previous performance levels upon retraining [3]. Athletes facing training interruptions may encounter risk factors that negatively impact performance, including declines in muscle strength and size, cardiovascular endurance, along with impairments in neuromuscular coordination and flexibility [3–5]. Additionally, prolonged periods of inactivity are associated with an elevated risk of musculoskeletal injuries upon return to training [6,7]. When the interruption period lasts for approximately four weeks, reductions are primarily observed in general endurance and muscular endurance performance, while muscle strength remains largely unaffected [8,9]. However, if training cessation exceeds four weeks, both muscle strength and endurance show significant declines [3]. Furthermore, reductions in muscle strength and size are influenced by the duration of the detraining period. Short-term detraining typically leads to minimal or no measurable change, whereas extended detraining results in a more rapid loss of muscle size compared to maximal strength [10–13]. Research indicates that after a period of training cessation, skinfold thickness, body fat mass, and percentage tend to increase, while fat-free mass and athletic performance decrease [5].

In elite wrestling, multiple performance components are required to execute consecutive high-intensity physical efforts. Flexibility, agility, reaction time, and the execution of wrestling-specific techniques form the foundation of these components. These attributes are essential for the effective application of strength during performance [14]. Following a 10-week training break in wrestlers, concentric and eccentric isokinetic strengths of the knee and shoulder flexor and extensor were assessed at various angular velocities. Significant decreases in isokinetic muscle strength were detected at angular velocities of 60°, 180°, and 300° [6]. Such reductions not only impair performance but also increase injury risk. In another study, a significant reduction in flexibility and agility was observed among collegiate wrestlers aged 18–21 years after a 10-week training cessation, regardless of changes in muscle mass percentage [14].

Several studies have shown that athletes can regain lost performance relatively quickly following short-term interruptions [15,16]. Moreover, Valenzuela et al. [17] emphasized that inappropriate or abrupt return-to-play protocols may lead to overuse injuries and compromise full recovery, underscoring the importance of gradual and individualized training resumption. The literature also suggests that muscle strength can be restored rapidly after short-term (~three-week) periods of inactivity. Yasuda et al. [16] reported that individuals with prior strength training experience were able to regain muscular strength through blood flow restriction training after a three-week detraining phase.

In addition to physical performance losses such as reduced strength and endurance, the autonomic nervous system (ANS)—a vital marker of general health and a determinant of athletic performance—may also be adversely affected by training cessation. Heart rate variability (HRV), which is frequently used as a non-invasive marker of ANS function, can exhibit significant deterioration following prolonged periods of inactivity. Sustained training interruptions are commonly associated with decreased HRV values, which may contribute to reductions in performance capacity [18].

In early 2020, the global outbreak of coronavirus disease (COVID-19) was declared a public health emergency of international concern. In response, social gatherings were restricted and training facilities were closed, causing widespread disruptions in athletic preparation. The impact on elite athletes was substantial, with ongoing championships suspended and major international events postponed.

Moreover, most elite athletes were compelled to train alone in isolated and often unsupervised environments, such as their homes. Upon the easing of lockdown restrictions, it was recommended that training programs be carefully structured and that athletes' performance and health parameters be closely monitored to ensure a safe return to competition. This approach helped minimize injury risk and supported a gradual recovery of physical condition [19,20].

Although several studies have examined the effects of prolonged inactivity on physical and physiological adaptations and performance in elite athletes, there is a lack of research specifically investigating these effects in elite wrestlers. In particular, little is known about the physiological recovery trajectory following 14 weeks of inactivity in this population. Furthermore, the duration required for elite wrestlers to fully regain optimal performance capacity in preparation for competition remains insufficiently understood [15,21,22].

The purpose of this study was to examine the changes in the physical and physiological parameters in elite wrestlers on the Turkish National Wrestling Team who underwent a 14-week period of limited and unstructured activity during the COVID-19 lockdown, followed by an 8-week retraining and competition phase.

## Materials and methods

### Experimental approach to the problem

This study is a longitudinal investigation that aims to examine the potential alterations in the physical, physiological, and performance metrics of elite senior national wrestlers competing in the Greco-Roman style. These wrestlers had not engaged in field-based training for a minimum of 14 weeks due to the Covid-19 epidemic. The study focuses on their performance outcomes and autonomous system responses following the resumption of field training. The recruitment period for the study was between 15.10.2020 and 15.02.2021.

 

During the 14-week COVID-19 lockdown period, information regarding the wrestlers' physical activity levels was obtained through self-reports collected by the Turkish National Wrestling Team coaches. Notably, one of the coaches involved in data collection is also listed as an author of this study, which provided close oversight and contributed to the reliability of the self-reported data. The athletes reported that they remained at home in accordance with pandemic restrictions and engaged only in basic home-based exercises, walking, and limited jogging. They did not participate in any structured or externally supervised training programs, and no specific dietary interventions were implemented. In the early phase of the lockdown, the wrestlers were predominantly inactive due to the uncertainty surrounding the duration of the restrictions. As the lockdown persisted, they gradually began to engage in limited individual exercises. Nevertheless, these activities lacked the structured intensity, volume, frequency, and duration characteristic of systematic athletic training.

HRV is a well-established non-invasive indicator of ANS regulation, specifically reflecting the balance between sympathetic and parasympathetic nervous system activities. Higher HRV is generally associated with increased parasympathetic (vagal) modulation and improved cardiovascular adaptability, whereas reduced HRV reflects sympathetic dominance and impaired autonomic flexibility [23]. In this study, HRV was measured to assess the wrestlers' general health and fitness status during the 8-week retraining period following the training cessation period, and also during the competition period that followed. HRV data were collected using a Polar H10 heart rate chest strap sensor.

To evaluate ANS through HRV, the following commonly used parameters were recorded: standard deviation of NN intervals (SDNN), root mean square of successive RR interval differences (RMSSD), peak frequency of the high-frequency band (HF), peak frequency of the low-frequency band (LF), and the LF/HF power ratio. HRV measurements, averaged over at least 10 days during the training cessation period, were obtained through 5-minute recordings taken while lying in bed immediately upon waking, each morning for 8 weeks after the retraining began.

The athletes' body weight, subcutaneous fat thickness, and body fat percentage were assessed on the first day they of training and subsequently at 2-week intervals throughout the retraining period. In addition, the athletes completed the Specific Wrestling Fitness Test (SWFT), a wrestling-specific assessment, on the first day of retraining and again after eight weeks. Blood lactate samples were collected immediately before, immediately after, and four minutes post-test to determine whether the SWFT was performed at the intended physiological intensity.

## Participants

This study included 20 male volunteer wrestlers (aged 21.75±3.11 years) who were members of the Turkish National Wrestling Team and had been training for at least 10 years. The wrestlers included in the study were elite-level athletes who were actively competing prior to the training cessation period. While organized competitions and group training sessions were cancelled during the COVID-19 pandemic, these athletes maintained limited, individualized training routines adapted to the imposed restrictions. However, these activities lacked the structured intensity, volume, frequency, and duration typically required for systematic athletic development. Two of the participants had previously won medals at the European and World Championships, and an additional seven athletes had achieved national and international medals in the year preceding the training cessation. Following the cessation period, one wrestler earned a silver medal at the World Championships and a gold medal the following year. Additionally, six athletes secured medals at European and World Championship events in the seasons following the cessation. The exceptional achievements of the participants, both prior to and following the training cessation period, clearly demonstrate their status as highly competitive elite wrestlers on the international stage.

The inclusion criteria for this study were as follows: being 19 years of age or older; being a national-level athlete with international competition experience; having similar lifestyle habits and sporting backgrounds; not taking any medication that could affect cardiac function; and having no known medical conditions that could impair cardiac or ANS function.

The exclusion criteria included: voluntary withdrawal from the study, failure to regularly attend scheduled measurement sessions, or sustaining an injury or a long-term illness during the training period.

All participants underwent a similar experience during the mandatory stay-at-home period, began retraining at the same time, and followed an identical training program throughout the retraining period. The study design and timeline are illustrated in Fig 1.

All participants provided written informed consent prior to participation. The study was conducted in accordance with the principles of the Declaration of Helsinki. This research was approved by the Akdeniz University Clinical Research Ethics Committee (KAEK-758/ 08.10.2020), and formal permission was obtained from the Wrestling Federation of Türkiye for the measurements.

### Anthropometric assessments

To evaluate the changes in the physiological characteristics of the participants, body weight (BW), body fat percentage (BFP), body muscle mass (BMM) and body mass index (BMI), measurements were assessed on the last day of training cessation period and 2-week intervals during the 8-week retraining period. Measurements were conducted using a bioelectrical impedance analysis device (Tanita Body Composition Analyzer Type SC-330).

BW, subcutaneous fat thickness (SFT), BFP measurements were measured immediately after training cessation, on the first day of the retraining, and at 2-week intervals thereafter. BMI was calculated using the formula of BMI = Weight (kg)/Height$^2$ (m$^2$). BMI values were reported for descriptive purposes only and were complemented with direct body composition measurements such as subcutaneous fat thickness (SFT) to better reflect the athletes' physical profiles. Height, weight, BMI, BFP values were recorded using the integrated printer on the body composition analyzer.

**Training program.** Due to the constraints imposed by the pandemic, the wrestlers unable to perform a structured training program during the long-term training cessation period. In accordance with the limitations of the COVID-19 conditions, training during this time consisted of basic strength exercises, individual running and walking routines and household-item assisted workouts that athletes could conduct at home. This period included only self-directed exercise, and no training program supervised by external trainers was implemented.

The retraining period consisted of standardized wrestling training sessions designed and monitored by the national team coaches, conducted at varying intensities over 8 weeks. Following the training cessation period (TCp), training intensity was progressively increased during the first 5 weeks (training period (Tp)). In the 6th and 7th week, training reached its peak (high intensity training period (HTp)). In the 8th week, leading into competition, training intensity was reduced to allow for tapering and recovery (form training period (FTp)). After the retraining period, wrestlers participated in an international competition (competition period (Cp)).

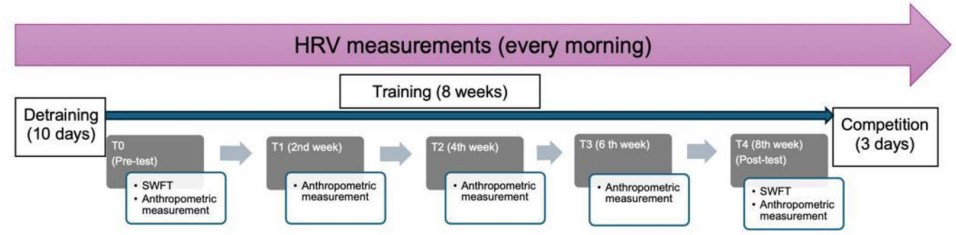

*HRV measured during the detraining for 10 days.
*Pre-tests were measured at the first day of training period
*HRV was measured every morning for 10 days in detraining period, eight weeks during the training period and 3 days during competition period.
*Post-tests were at the end of the T4 period, two days before competition.

**Fig 1. Trial Design (Approximately Here).**

Daily training logs were maintained to monitor compliance, and these records were regularly reviewed by the researcher (B.D.). Wrestlers were required to attend at least 90% of the training sessions. The weekly breakdown of the 8-week training program is provided in Table 1.

## HRV measurements

HRV was measured daily throughout the training cessation phase, which lasted 10 days prior to the start of the retraining sessions. This phase followed an 8-week training program and a 4-day competition period. Participants were provided a Polar H10 heart rate chest strap sensor (Polar Electro Oy, Kempele, Finland) with a sampling rate of 1000 Hz. A mobile application, Elite HRV, was used to record, store, and export the data.

Prior to data collection, participants were instructed on how to use their assigned HR monitor and smartphone application (Elite HRV). They were directed to measure their resting heart rate for 5 minutes each morning upon waking, while remaining in a supine position. The measurements involved evaluating the R-R intervals, with statistical analyses including SDNN and RMSSD, as well as frequency-domain analyses such as LF, HF, and LF/HF ratio [24]. Mean HRV values were calculated separately for each training phase (TCp, Tp, HTp, FTp, Cp), categorized based on training intensity.

## Specific Wrestling Fitness Test (SWFT)

The SWFT was used to evaluate physical fitness capacity of the wrestlers. Participants performed the test before and after the 8-week retraining period. Blood lactate concentrations were measured before, immediately after, and 4 minutes after the SWFT using fingertip samples and a portable lactate analyzer (EDGE Blood Lactate Analyzer, Woodley Equipment Company, Lancashire).

The SWFT consists of maximum wrestling dummy throws within a timed protocol. As wrestling is a weight-classified sport, participants used dummies corresponding to their weight class:

**Table 1. The weekly content of the 8-week training program.**

| Week 1 | Week 2 | Week 3 | Week 4 |
|---|---|---|---|
| **Total exercise session:** 8<br>**Exercise types:**<br>**Type 1-)** 3 BST sessions (%60–70 of 1RM)<br>**Type 2-)** 3 WFT (1,5 hours average of each)<br>**Type 3-)** 2 AE sessions (1 hour running, %50–60 of HRmax, sauna) | **Total exercise session:** 10<br>**Exercise types:**<br>**Type 1-)** 4 BST sessions (%60–75 of 1RM)<br>**Type 2-)** 4 WFT sessions (1,5 hours average of each)<br>**Type 3-)** 2 AE (1 hour running, %60–70 of HRmax, sauna) | **Total exercise session:** 11<br>**Exercise types:**<br>**Type 1-)** 4 BST sessions (%70–75 of 1RM)<br>**Type 2-)** 5 WFT sessions (2 hours average of each)<br>**Type 3-)** 1 AE session (45 minutes running, %70–75 of HRmax, sauna)<br>**Type 4-)** 1 session core exercises | **Total exercise session:** 11<br>**Exercise types:**<br>**Type 1-)** 3 BST sessions (%70–75 of 1RM)<br>**Type 2-)** 6 WFT sessions (2 hours average of each)<br>**Type 3-)** 1 HIIT running session (30 minutes, %70–80 of HRmax)<br>**Type 4-)** 1 session core exercises |
| Week 5 | Week 6 | Week 7 | Week 8 |
| **Total exercise session:** 10<br>**Exercise types:**<br>**Type 1-)** 3 BST sessions (%75 of 1RM)<br>**Type 2-)** 5 WFT sessions (2 hours average of each)<br>**Type 3-)** 1 AE session (1 hour running, %60–70 of HRmax, sauna)<br>**Type 4-)** 1 session core exercises | **Total exercise session:** 10<br>**Exercise types:**<br>**Type 1-)** 3 SMT session (%50–60 of 1RM)<br>**Type 2-)** 5 WFT sessions (1,5 hours average of each)<br>**Type 3-)** 1 HIIT running session (30 minutes, %70–75 of HRmax, sauna)<br>**Type 4-)** 1 HIIT running session (20 minutes, 80–90 of HRmax) | **Total exercise session:** 8<br>**Exercise types:**<br>**Type 1-)** 2 EST session (%80–90 of 1RM)<br>**Type 2-)** 5 WFT sessions (1 hour average of each)<br>**Type 3-)** 1 HIIT running session (20 minutes, 80–90 of HRmax, restitution, sauna) | **Total exercise session:** 6<br>**Exercise types:**<br>**Type 1-)** 2 EST session (%90–100 of 1RM)<br>**Type 2-)** 3 WFT sessions (30–45 minutes average of each)<br>**Type 3-)** 1 restitution trainings (30 minutes, sauna) |

BST: Basic strength training, WFT: Wrestling field training, AE: Aerobic exercise, SMT: Strength maintenance training, HIIT: High intensity interval training, EST: Explosive strength training, HRmax: Maximum heart rate, 1RM: 1 Repetition maximum.

- 22 kg dummy for athletes weighing up to 74.9 kg.

- 27 kg dummy for those weighing between 75.0–89.9 kg.

Each participant performed three sets of 30-second dummy throws, separated by 20-second rest intervals. After the start signal, participants attempted to throw the dummy as many times as possible within the allotted time using maximum effort [25].

The total number of throws completed across the three sets was recorded as the final test score. Prior to each test, athletes completed a 15-minute general warm-up, followed by a 5-minute specific warm-up involving wrestling dummy throws. A visual illustration of the dummy used in the SWFT is provided in Supplementary Fig 1.

### Statistical analyses

Several statistical methods were employed to test for normality, including assessments of kurtosis and skewness, histogram visualizations, Q-Q plots, and boxplots, followed by the Shapiro-Wilk test. For comparing pre- and post-test values within groups, either the Dependent Samples t-test or the Wilcoxon signed-rank test was used, depending on data distribution. To analyze repeated measurements over time, Repeated Measures ANOVA was conducted. All analyses were performed using Jamovi software (version 2.3.18).

### Results

Anthropometric assessments were conducted biweekly throughout the training period. Subcutaneous fat thickness was measured at the triceps, abdomen, subscapular, supraspinale, pectoral, midaxillary, and quadriceps regions using the skinfold method. Additionally, body fat percentage and body muscle mass were assessed using the bioelectrical impedance analysis (BIA) method. Statistically significant improvements were observed in both BFP and BMM parameters at each subsequent measurement point compared to the previous one. (Table 2).

There was a significant time x group interaction effect in subcutaneous fat thickness measurements. Notable reductions were observed across all assessed sites during the 8-week retraining period (Fig 2). Triceps skinfold thickness decreased significantly at each measurement point compared to the previous one ($p < 0.01$). Similarly, abdominal ($p < 0.001$), supraspinale ($p < 0.001$), midaxillary ($p < 0.01$), subscapular ($p < 0.001$), and quadriceps ($p < 0.01$) regions exhibited statistically significant reductions at every time point. In the pectoral region, a significant reduction was observed at week 2 ($p < 0.01$) and week 4 ($p < 0.05$); however, no further significant changes were observed thereafter. These results collectively

**Table 2. Biweekly Changes in Body Weight, Body Fat Percentage, Muscle Mass, and BMI During the Retraining Period.**

| n = 20 | W0 | W2 | W4 | W6 | W8 | F | p | η2 |
|---|---|---|---|---|---|---|---|---|
| **BW** (kg) | 76.49 ± 12.27 | 75.29 ± 11.58* | 74.9 ± 11.33 | 75.08 ± 11.37 | 75.47 ± 11.61 | 6.918 | 0.000 | .27 |
| | 70;82 | 70;81 | 70;80 | 70;80 | 70;81 | | | |
| **BMI** (kg/m²) | 25.11 ± 2.74 | 24.73 2.56** | 24.60 ± 2.48 | 24.67 ± 2.46 | 24.78 ± 2.44 | 7.237 | 0.000 | .28 |
| | 24;26 | 24;26 | 23;26 | 24;26 | 24;26 | | | |
| **BFP** (%) | 12.38 ± 4.50 | 11.25 ± 4.19*** | 10.77 ± 3.96*** | 10.12 ± 3.77*** | 9.66 ± 3.66** | 44.751 | 0.000 | .70 |
| | 10;15 | 9;13 | 9;13 | 8;12 | 8;11 | | | |
| **BMM** (kg) | 62.51 ± 7.86 | 62.90 ± 7.95** | 63.25 ± 8.11*** | 64.08 ± 8.37*** | 65.00 ± 8.80*** | 39.678 | 0.000 | .68 |
| | 59;66 | 59;67 | 59;67 | 60;68 | 61;69 | | | |

Data in the table; presented as mean ± standard deviation (Mean ± SD) and 95% confidence interval (5%; 95% CI). *$p < 0.05$, **$p < 0.01$ ***$p < 0.001$ compared to the previous measurement. Statistical analysis was performed using repeated measures ANOVA followed by Bonferroni post-hoc tests. W0: pre-training period; W2: 2 weeks post-training, period; W4: 4 weeks post-training period; W6: 6 weeks post-training period; W8: 8 weeks post-training period; BW: body weight; BMI: body mass index; BFP: body fat percentage; BMM: body muscle mass; η2: partial eta squared (effect size).

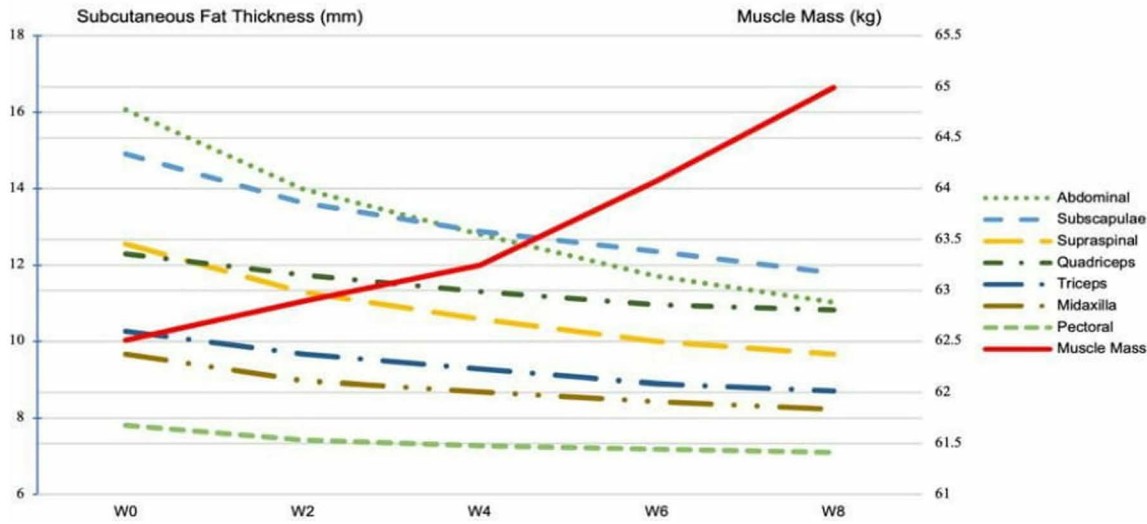

**Fig 2. Changes in subcutaneous fat thickness and muscle mass (Approximately here).**

suggest a progressive improvement in body composition during the retraining phase, as evidenced by the consistent decrease in subcutaneous fat thickness across the majority of the measured anatomical sites.

SWFT scores increased significantly following the 8-week training period (Table 3). Despite the improvement in performance, post-test blood lactate levels were lower compared to pre-training values, indicating enhanced anaerobic efficiency and lactate clearance capacity. (Fig 3).

Table 4 and Fig 3 present the HRV variables of the participants. A significant time x group interaction was observed for HRV scores, SDNN, RMSSD, LF, LF/HF values. HRV, SDNN, LF values were significantly higher during the FTp compared to both TCp and Cp.

## Discussion

This study aimed to investigate the effects of an 8-week retraining program on various physical, physiological, and athletic performance parameters, as well as on ANS tone—specifically examining the balance between the sympathetic and

**Table 3. SWFT scores and Lactate Levels.**

| (n = 20) | T0 | T4 | t | p | E |
|---|---|---|---|---|---|
| **SWFT (throws)** | 31.40 ± 2.91 30;33 | 37.40 ± 3.22 36;39 | 10.104 | 0.000 | 2.23 |
| | | | F | p | η2 |
| **LApre (mmol/L)** | 1.18 ± 0.33 1;1 | 1.03 ± 0.29 1;1 | 724.028 | 0.000 | 0.950 |
| **LApost (mmol/L)** | 5.21 ± 2.15 4;6 | 4.88 ± 1.55 4;6 | | | |
| **LA4post (mmol/L)** | 10.45 ± 2.84 9;12 | 8.10 ± 1.55 7;9 | | | |

Data in the table; presented as mean ± standard deviation (Mean ± SD) and 95% confidence interval (5%; 95% CI).

W0: pre-training period; W2: 2 weeks post-training, period; W4: 4 weeks post-training period; W6: 6 weeks post-training period; W8: 8 weeks post-training period; SWFT: specific wrestling fitness test; LApre: Lactate levels pre SWFT; LApost: Lactate levels immediately post SWFT; LA4post: Lactate levels post 4 minutes SWFT; E: effect size, η2: partial eta squared.

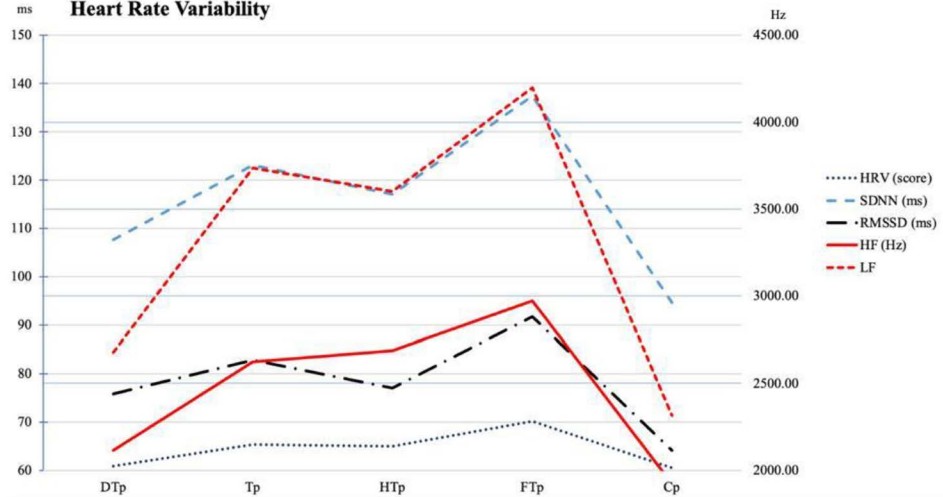

**Fig 3. HRV values (Approximately here).**

**Table 4. HRV Values.**

| (n=20) | DTp | Tp | HTp | FTp | Cp | F | p | η2 |
|---|---|---|---|---|---|---|---|---|
| **HRV** | 60.80±8.83 | 65.37±6.38 | 64.97±4.63 | 70.12±8.91 | 60.45±6.94 | 11.384 | 0.002 | 0.487 |
| | 55;66 | 62;69 | 62;68 | 65;76 | 56;65 | | | |
| **SDNN** | 110.01±55.03 | 122.76±35.50 | 116.35±35.02 | 131.43±37.19 | 94.60±25.63 | 5.330 | 0.001 | 0.308 |
| | 77;143 | 101;144 | 95;138 | 109;154 | 79;110 | | | |
| **RMSSD** | 73.49±40.63 | 80.66±33.85 | 75.22±21.86 | 86.98±27.05 | 64.06±14.91 | 2.719 | 0.040 | 0.185 |
| | 49;98 | 60;101 | 62;88 | 71;103 | 55;73 | | | |
| **LF** | 2456±1049 | 3745±1638 | 3941±1818 | 3979±1775 | 2314±1137 | 6.191 | 0.000 | 0.340 |
| | 1821;3090 | 2755;4735 | 2392;4590 | 2906;5052 | 1626;3001 | | | |
| **HF** | 2172±1161 | 2732±1487 | 2757±1804 | 2946±1583 | 2923±1691 | 2.190 | 0.130 | 0.154 |
| | 1471;2874 | 1833;3630 | 1667;3848 | 1989;3901 | 907;2939 | | | |
| **LF/HF** | 2.65±0.86 | 4.04±2.11 | 3.79±1.71 | 4.08±2.18 | 2.79±1.24 | 3.162 | 0.022 | 0.209 |
| | 2;3 | 3;5 | 3;5 | 3;5 | 2;4 | | | |

Data in the table; presented as mean±standard deviation (Mean±SD) and 95% confidence interval (5%; 95% CI).

DTp:detraining period; Tp: training period; HTp: high intensity training period; FTp: form training period; Cp:competition period; HRV: heart rate variable score; SDNN: standard deviation of NN intervals, RMSSD: root mean square of successive RR interval differences, HF: peak frequency of the high-frequency band, LF: peak frequency of the low-frequency band (), and LF/HF: ratio of LF-to-HF power, η2: partial eta squared (effect size).

parasympathetic branches—in elite national team wrestlers following a 14-week period of training cessation. The findings revealed significant improvements in body composition, athletic performance, and HRV values—an established indicator of ANS function—following the 8-week retraining period.

Training cessation, or detraining, refers to the partial or complete loss of physiological adaptations previously gained through training, occurring when physical activity is reduced or stopped [3]. Short-term detraining has been reported to result in marked increases in body fat mass (+21%) and reductions in muscle mass (~4%) in elite athletes, potentially impairing performance and agility [26]. When detraining extends beyond four weeks, its negative impact on body fat percentage, muscle mass, and overall performance becomes more pronounced. If prolonged further, it may lead to the partial

or total loss of training-induced adaptations. The magnitude of these changes varies depending on the individual's initial fitness level and the duration of the detraining period. [27].

Previous studies have shown that body fat mass, body fat percentage, and skinfold thickness tend to increase, while fat-free mass and athletic performance decrease following a period of training cessation [5,28]. In wrestling, maintaining optimal body composition—characterized by a low body fat percentage—is particularly crucial, as athletes are matched against opponents of similar body weight before each bout. A lower fat mass (FM) percentage is considered advantageous, while higher fat-free mass (FFM) is recognized as a key determinant of anaerobic performance in wrestlers. Therefore, balancing fat reduction and muscle preservation is essential for optimizing competitive readiness and athletic capability [29].

Although elite athletes may experience short-term benefits, such as super-compensation and improved recovery following a temporary reduction in training, the long-term consequences of training cessation are detrimental to their overall conditioning and performance. Elite wrestlers typically engage in three to six hours of intensive training per day to optimize their athletic capabilities. As such, their training programs must consistently target the development of strength, flexibility, and anaerobic power to maintain and enhance performance levels [30,31]. Under normal conditions, wrestlers take short breaks or reduce training volume and intensity for no more than three weeks after the competitive season. However, the COVID-19 pandemic forced elite athletes into an unusually prolonged period of inactivity. The pandemic led to widespread disruptions in training, limiting access to facilities and structured programs, and resulting in extended periods of detraining that negatively affected physical performance and increased the risk of injury [32,33]. The adverse effects of prolonged training cessation include impairments in their physical condition and athletic performance, increases in body fat percentage, heightened fatigue, a greater risk of injury, and longer recovery times [19,20]. Even inactivity periods as short as 2–4 weeks can induce significant detraining effects, such as reductions in maximal oxygen uptake, muscle strength, neuromuscular coordination, and ventilatory efficiency. These physiological declines can increase susceptibility to injury upon return if training intensity is ramped up too quickly [3,34].

Grazioli et al. [35] reported that an 8-week quarantine period negatively impacted football players' physical performance -specifically power, sprinting, body mass, and body fat mass- and emphasized the need for longer recovery timelines than those associated with regular training. Similarly, Valenzuela et al. [17] highlighted that following long-term inactivity such as during the COVID-19 lockdown, athletes require carefully structured retraining programs lasting at least 6–8 weeks to restore physiological and performance capacities.

Regardless of the cause of training interruption, previous studies have shown that the longer the inactivity period, the more pronounced the declines in endurance, strength, and neuromuscular coordination. These findings underscore the necessity of structured and individualized retraining strategies for athletes returning after extended breaks [3]. Sudden increases in training volume and intensity following confinement can overload physiological systems and elevate injury risk. Conversely, appropriate retraining approaches can minimize injury risk, prevent long-term losses in muscle size and strength, and support a more reliable return to optimal performance. Structured, individualized retraining programs are especially critical for elite wrestlers after prolonged inactivity. To mitigate performance declines and safely return to competition, athletes must implement evidence-based strategies, including progressive training loads, baseline physical assessments, and regular monitoring using tools such as HRV and perceived exertion scales. [15,16,36,37].

In our study, we aimed to determine the physical and physiological adaptations that occur following an eight-week retraining program implemented after a prolonged 14-week period of inactivity. During this retraining period, a clear and progressive improvement was observed in body fat percentage, muscle mass, and athletic performance. However, as noted by Grazioli et al. [35] there is currently no research in the literature that allows for direct comparison regarding the time required for such improvements. Nonetheless, our findings suggest that elite wrestlers can achieve substantial gains in body composition and athletic performance within just eight weeks of training, even after an unusually extended period of inactivity.

 

The development of anaerobic endurance is crucial for explosive sports like wrestling. The SWFT is a widely used performance assessment tool to measure wrestlers' anaerobic endurance and overall physical fitness. The total number of wrestling dummy throws serves as an indicator of immediate physical fitness and anaerobic capacity [25]. Anaerobic endurance, maximal strength, and dynamic balance are fundamental components of wrestling performance. The rapid recovery of these parameters following a period of detraining is essential for maintaining explosive movement patterns and technical efficiency during competition. Although there is currently no study specifically examining the combined effects of training cessation and retraining periods, previous research indicates that wrestlers can improve their SWFT score after six weeks of multi-component training [30,38]. In combat sports such as wrestling, decision-making skills and technical proficiency require practice against live opponents. Following 4–12 weeks of reduced training (~20–40% of normal load), a gradual return involving general physical preparation and progressive specific skill training over an estimated 3–5 weeks has been recommended to minimize injury risks [39].

In our study, it was observed that elite-level wrestlers significantly improved their SWFT scores by the end of the 8-week retraining period. Moreover, although the SWFT scores increased significantly, post-test lactate levels were notably lower during the retraining phase (8.10±1.55 vs. 10.45±2.84 mmol/L). This reduction in lactate levels, despite an increase in the number of throws performed, indicates improvements both lactate threshold and lactate clearance capacity.

Furthermore, a key finding of our study is the relationship between HRV and both training intensity and competition duration. While consistent long-term training can enhance HRV in athletes, excessively intense training may result in significant daily HRV reductions, and lower morning HRV values may reflect a decline in performance capacity. Athletes experiencing overtraining syndrome often exhibit markedly reduced HRV [40]. In addition, training cessation may also lead to a reduction in HRV. According to Ruiz-Navarro et al. [41], after five weeks of training cessation, all vagal-related HRV measures recorded before and after exercise showed a decline, indicating a deterioration in the swimmers' physical condition. The state of the ANS is influenced by accumulated physical fatigue resulting from increased training loads. Therefore, monitoring HRV can serve as an effective tool for assessing an athlete's training status and overall performance and may also be used to help prevent overtraining [40–42]. Individually tailored training load management can maximize athletic performance gains, whereas both excessive and insufficient training loads may result in detraining and accumulated fatigue. HRV monitoring has been recognized as an effective method for evaluating an athlete's physiological status in relation to sports performance [17].

In our study, HRV values were monitored throughout the eight-week retraining period and the subsequent competition phase. The data were evaluated by analyzing average HRV values across specific time intervals, categorized based on training load intensity. Overall, HRV demonstrated a progressive improvement during the retraining phase but showed a modest decline during periods of high training intensity. Notably, during the tapering phase – when training intensity was reduced- HRV values reached their peak levels. Valenzuela et al. [17] suggested that the COVID-19 lockdown could have a detrimental effect on HRV in elite athletes. Similarly, Gamelin et al. [43] reported that an eight-week period of training cessation may reverse cardiovascular autonomic adaptations previously achieved through training. On a more optimistic note, Valenzuela et al. [17] reported that elite athletes tend to achieve full recovery following 6–8 weeks of retraining, which aligns with the findings of our study. However, in our study, although HRV values progressively improved during the retraining phase, they declined again during the subsequent competition period—reaching levels comparable to those observed during the initial training cessation. This decline in HRV suggests that athletes may have experienced increased physiological stress and fatigue during competition, potentially impairing autonomic recovery. Several studies [44,45] have demonstrated that stress and anxiety negatively influence the ANS during competitive periods, resulting in decreased HRV values. Additionally, research on recovery following injury or forced inactivity consistently shows that restoring muscle mass, cardiovascular capacity, and neuromuscular control require progressive retraining over several weeks to return to baseline levels [19]. It has also been reported that lockdown conditions caused measurable declines in aerobic fitness and neuromuscular performance among elite athletes, underscoring the universal vulnerability of this population to detraining effects, regardless of the reason for inactivity [46].

Collectively, these findings highlight the value of HRV monitoring as a non-invasive tool for assessing athletes' physiological responses to stress and anxiety during competition. Such insights can guide the development of more effective training and recovery strategies. Furthermore, implementing interventions aimed at reducing psychological stress may help maintain optimal HRV levels and support improved performance outcomes in competitive settings.

The most significant limitation of this study is the absence of a control group. After the isolation period ended, all athletes were allowed and required to resume training simultaneously. In elite athletic populations, establishing a control group is particularly challenging, as training cessation or alterations to the standard program are generally not accepted by coaches or athletes.

Another limitation is the lack of pre-pandemic or during-pandemic baseline measurements. Due to the sudden and unforeseen implementation of pandemic-related restrictions, initial assessments could not be conducted in accordance with mandatory social distancing regulations. As a result, baseline measurements were taken at the beginning of the retraining phase, and subsequent changes were tracked over time. Additionally, the study employed a convenience sampling method, as access to participants was severely restricted during the COVID-19 pandemic. Although this approach was the most practical under the given circumstances, it may limit the generalizability of the findings. Future studies are encouraged to adopt randomized or stratified sampling methods to enhance sample representativeness and strengthen the robustness of the conclusions.

Elite athletes possess a remarkable capacity to regain both athletic performance and body composition following periods of training cessation. HRV, a key marker of ANS function, fluctuates significantly in response to changes in training load and competition stress. These fluctuations underscore the importance of HRV monitoring for optimizing performance and recovery strategies.

A thorough understanding of these physiological responses can empower coaches and athletes to make more informed decisions, facilitate targeted adaptations, and ultimately achieve higher levels of athletic success. Based on the findings of our study, it is recommended that wrestlers returning from prolonged training cessation follow retraining programs that emphasize anaerobic conditioning, technical-tactical drills performed under fatigue, and gradual progression of training loads to minimize injury risk and accelerate the restoration of competitive readiness.

## Conclusions

As a result of eight-week retraining period, a clear and progressive decrease in body fat percentage and an increase in muscle mass were observed. Following this training period, the Specific Wrestling Fitness Test (SWFT), which is used to assess wrestlers' physical fitness capacity, showed significant improvements in performance scores.

Throughout the retraining period, HRV values exhibited a gradual upward trend, reflecting improved autonomic balance. However, a slight decline in HRV was noted during phases of intensive training. During the tapering phase, characterized by reduced training intensity, HRV values reached their highest levels.

These findings indicate that HRV is highly sensitive to variations in training load and competitive stress. Notably, HRV values measured during the competition phase were similar to those observed during the non-training period, suggesting a potential physiological strain associated with competitive demands.

## Supporting information

**S1. Supplementary Figure.**
(TIF)

## Acknowledgments

The authors would like to thank the participants for volunteering their time and the Türkiye National Greco-Roman Wrestling Team.

The English language, grammar, and overall clarity of this manuscript were reviewed and edited with the assistance of an advanced AI language model (ChatGPT-4o, OpenAI). Following the AI-supported editing, final revisions and adaptations were carefully made by the authors to ensure that the manuscript meets international academic writing standards and accurately reflects the intended scientific meaning.

## Author contributions

**Conceptualization:** Tuba Melekoglu.

**Data curation:** Burhan DEMİRKIRAN, Ali IŞIN, Yılmaz SUNGUR, Tuba Melekoglu.

**Formal analysis:** Tuba Melekoglu.

**Funding acquisition:** Tuba Melekoglu.

**Investigation:** Burhan DEMİRKIRAN.

**Methodology:** Burhan DEMİRKIRAN, Tuba Melekoglu.

**Supervision:** Tuba Melekoglu.

**Writing – original draft:** Burhan DEMİRKIRAN, Ali IŞIN, Yılmaz SUNGUR, Tuba Melekoglu.

**Writing – review & editing:** Tuba Melekoglu.

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
