## [Decision Letter · Decision Letter 0]

Dear Dr. Melekoglu,

Thank you for submitting your manuscript to PLOS ONE. After careful consideration, we feel that it has merit but does not fully meet PLOS ONE’s publication criteria as it currently stands. Therefore, we invite you to submit a revised version of the manuscript that addresses the points raised during the review process.

We look forward to receiving your revised manuscript.

Kind regards,

Ozkan Isik

Academic Editor

PLOS ONE

Journal Requirements:

2. Please amend your list of authors on the manuscript to ensure that each author is linked to an affiliation. Authors’ affiliations should reflect the institution where the work was done (if authors moved subsequently, you can also list the new affiliation stating “current affiliation:….” as necessary).

Reviewers' comments:

Reviewer's Responses to Questions

**Comments to the Author**

1. Is the manuscript technically sound, and do the data support the conclusions?

Reviewer #1: Yes

Reviewer #2: Partly

2. Has the statistical analysis been performed appropriately and rigorously?

Reviewer #1: Yes

Reviewer #2: I Don't Know

3. Have the authors made all data underlying the findings in their manuscript fully available?

Reviewer #1: Yes

Reviewer #2: Yes

4. Is the manuscript presented in an intelligible fashion and written in standard English?

Reviewer #1: Yes

Reviewer #2: Yes

Reviewer #1: It is thought that the relevant research will contribute to the literature, especially in terms of emphasizing the correct loading techniques during the adaptation process of athletes who have been away from training due to various reasons. In addition, it is observed that the relevant research is in an important place due to the fact that injuries are frequently experienced and athletes are away from the fields for a long time, such as wrestling. For this reason, it is anticipated that the article related to the mentioned minor revisions will be more understandable and will emphasize the purpose more clearly. In this context, if the mentioned corrections are made, the relevant article is suitable for publication in PLOS One.

Abstract

-The problems and risk factors that athletes encounter after the training stop period should be expressed more clearly and in detail.

-The reason why it is important to use the right method after the training stop period should be expressed more clearly.

-It should be stated which sporting performance parameters the training stop period can affect in wrestling in particular.

-Since the most striking aspect of the research is the importance of correct loading and quickly reaching optimum performance after a long training stop period, this issue should be emphasized more.

Materials and Methods

-More detailed information should be provided about which period the athletes are in before the training stop period (in-season, end of season or rest period).

-The training program applied to the athletes for 8 weeks should be expressed in more detail.

-The movements and their durations preferred in the training program applied to the athletes should be clearly expressed in tables.

-It should be stated more clearly how the athletes follow up whether or not they apply the specified training during the 8-week period.

-It is recommended to add the visual of the wrestling mannequin used in SWFT to increase the explanatory power of the test used.

Results

-The statistical methods applied and the analyses included in the article are explanatory and appropriate for the purpose.

Discussion

-Since the main purpose of the research is to ensure fast and accurate recovery after a long period without training, the importance of this subject should be emphasized more.

-Although there is no similar research on this subject during the Covid-19 period, a comparison should be made in the discussion section with studies that express how many weeks and which methods are applied to athletes, especially after injuries or when the athlete is away from the field for a long time.

-Although the relevant research was conducted in a specific process such as Covid-19, the findings obtained are very important for athletes who have to stop training for a long time. Therefore, more space should be given to similar studies suitable for the specified athlete profile.

-In the relevant studies, the importance of the parameters examined for the wrestling branch and how these parameters can affect performance with rapid recovery should be included in the foresight and similar studies. -More suggestions should be made regarding the results of the research, specific to the wrestling branch.

Reviewer #2: Page 12, line 25 – so you have mentioned that HRV is the key variable observed and then you don’t mention the effects of detraining and retraining on it?

Line 14, line 62 – you mean the prevention of social gatherings?

Page 14, line 69 – why do we assume that athletes were not active during COVID pandemic? They were competing okay, but many of them remained active in the conditions they were given.

Page 15, line 78 – again… How do you know they were unactive? Did you track their behavior somehow?

Page 15, line 95 – indicator of what specifically within the ANS? Provide details.

Page 16, line 106 – which tools have you used to measure these?

Line 18, line 144 – BMI is often times not a relevant measure for athletes, especially for wrestlers many of whom are short and have large muscle mass. Reconsider involving this measurement.

Page 19, line 175 – what is SDNN and RMSSD? Explain before using abbreviations.

Page 20, line 195 – this is repetitive… you’ve already said when the measurements were taken

Page 21, 208 -again repetitive, you’ve already stated this about your sample

Page 21, line 209-10 – which measurements specifically?

Your study is not cross-sectional, but rather longitudinal because you have several measurements over a period of time.

Another major limitation to your study is convenience sampling.

You are missing a table that describes your participants.

Major revision is needed.

A native English speaking person reading and adjusting this text would be of great value.

**Do you want your identity to be public for this peer review?** For information about this choice, including consent withdrawal, please see our Privacy Policy

Reviewer #1: No

Reviewer #2: No

---

## [Author Response · Author response to Decision Letter 1]

13 May 2025

Reviewer #1:

We sincerely thank you for your valuable comments and suggestions. We have carefully addressed each of your points and revised the manuscript accordingly. Below, we provide a detailed, point-by-point response. The authors’ responses are written in italics. Where appropriate, changes made in the manuscript have been highlighted in yellow.

Suggestion: The problems and risk factors that athletes encounter after the training stop period should be expressed more clearly and in detail.

Response: In the Introduction and Abstract sections, we added information about decreases in muscle strength, cardiovascular endurance, flexibility, neuromuscular coordination, and the increased injury risk associated with training cessation. The following paragraph has been added to the manuscript:

“Athletes facing training interruptions may encounter risk factors that negatively impact performance, including declines in muscle strength, muscle size, cardiovascular endurance, as well as impairments in neuromuscular coordination and flexibility. Additionally, prolonged periods of inactivity are associated with an elevated risk of musculoskeletal injuries upon return to training”

Suggestion: The reason why it is important to use the right method after the training stop period should be expressed more clearly.

Response: The Abstract was revised to emphasize that appropriate retraining methods are critical for minimizing injury risk and for rapidly regaining optimal performance. The following paragraph has been added to the manuscript:

“Proper training methods can minimize the risk of injury, prevent long-term losses in muscle size and strength, and facilitate a faster and more reliable return to optimal fitness and performance levels. Structured and individualized retraining programs are crucial for minimizing injury risk and accelerating performance recovery in elite wrestlers following prolonged periods of training cessation. Given the detrimental effects of training cessation on athletic performance, it is essential for athletes to implement appropriate strategies to mitigate performance declines. Several studies have indicated that athletes are capable of regaining lost performance relatively quickly after short-term interruptions (Stokes et al., 2020; Yasuda et al., 2015). Moreover, Valenzuela et al. (Valenzuela et al., 2021) emphasized that inappropriate or abrupt return-to-play protocols might lead to overuse injuries and hinder full recovery, highlighting the critical need for gradual and individualized training resumption.”

Suggestion: It should be stated which sporting performance parameters the training stop period can affect in wrestling in particular.

Response: Wrestling-specific performance parameters, including muscle strength, flexibility, agility, and reaction time, were detailed in the Abstract and Introduction. The following paragraph has been added to the manuscript:

“In elite wrestling, multiple performance components are required to perform consecutive high-intensity physical activities. Flexibility, agility, reaction time, and the execution of wrestling-specific techniques form the foundation of these components. These elements are essential for the correct and efficient application of physical strength during performance (Hedya, 2016). Following a 10-week training break in wrestlers, the concentric and eccentric isokinetic strengths of the knee and shoulder flexor and extensor muscles were evaluated at different angular velocities. Significant decreases in isokinetic muscle strength were detected at angular velocities of 60°, 180°, and 300° (Noorbakhsh et al., 2025). Such declines in muscle strength not only contribute to performance losses but also represent a major risk factor for injuries. In another study, a significant reduction in flexibility and agility was observed among collegiate wrestlers aged 18–21 years after a 10-week training cessation, independent of changes in muscle mass percentage (Hedya, 2016).”

Suggestion: Since the most striking aspect of the research is the importance of correct loading and quickly reaching optimum performance after a long training stop period, this issue should be emphasized more.

Response: We revised both the Abstract and the Discussion sections to highlight the importance of correct loading strategies in achieving rapid and safe performance recovery.

In Abstract:

“Obtaining a comprehensive understanding of these changes can enable coaches and athletes to make accurate decisions in order to optimize training adaptations and attain overall athletic success.”

“Furthermore, over a period of eight weeks following a long non-training period, significant improvements in athletes’ body fat, muscle mass and wrestling performance can be achieved along with training.”

In Discussion:

“The proper methods can minimize the risks of injury, prevent long-term loss of performance such as muscle size and strength and allow a faster and more reliable return to optimum fitness and conditions. Structured and individualized retraining programs appear crucial in minimizing the risks of injury and accelerating performance recovery in elite wrestlers after prolonged training cessation.”

Materials and Methods

Suggestion: More detailed information should be provided about which period the athletes are in before the training stop period (in-season, end of season or rest period).

Response: It was clarified that the athletes were in their in-season period before the onset of the training cessation.

“The wrestlers in the research group consisted of elite level athletes who were in-season before the training cessation period.”

Suggestion: The training program applied to the athletes for 8 weeks should be expressed in more detail.

Response: We provided a detailed description of the 8-week training program and included a week-by-week training content in Table 1.

Suggestion: The movements and their durations preferred in the training program applied to the athletes should be clearly expressed in tables.

Response: A comprehensive table (Table 1) was added, detailing the exercise types, session frequency, intensity, and weekly structure.

Suggestion: It should be stated more clearly how the athletes follow up whether or not they apply the specified training during the 8-week period.

Response: We added information about daily training records being kept and reviewed by a researcher (B.D.), ensuring at least 90% session compliance.

“Daily training records were kept for the athlete to follow the training, and these records were regularly reviewed by the researcher (B.D.). Wrestlers are required to perform 90 % of the training sessions.”

Suggestion: It is recommended to add the visual of the wrestling mannequin used in SWFT to increase the explanatory power of the test used.

Response: In accordance with the recommendation, we have added visuals of the wrestling mannequins used in the Specific Wrestling Fitness Test (SWFT) to enhance the explanatory power and clarity of the method section. The images have been provided as a supplementary file and are referenced appropriately within the manuscript.

Results

Suggestion: The statistical methods applied and the analyses included in the article are explanatory and appropriate for the purpose.

Response: No change needed. Statistical analyses were already appropriate, but minor clarifications regarding repeated measures ANOVA and other tests were added.

Discussion

Suggestion: Since the main purpose of the research is to ensure fast and accurate recovery after a long period without training, the importance of this subject should be emphasized more.

Response: We emphasized this point in the opening and closing sections of the Discussion.

“The proper methods can minimize the risks of injury, prevent long-term loss of performance such as muscle size and strength and allow a faster and more reliable return to optimum fitness and conditions. Structured and individualized retraining programs appear crucial in minimizing the risks of injury and accelerating performance recovery in elite wrestlers after prolonged training cessation. Due to the negative impact of training cessation on athletic performance, it is essential for athletes to adopt appropriate strategies to counteract the decline in performance. Indeed, several studies have emphasized that athletes are able to regain lost performance relatively quickly after short-term cessations [1, 2]. Moreover, Valenzuela et al. [3] emphasized that inappropriate or abrupt return-to-play protocols might lead to overuse injuries and hinder full recovery, highlighting the critical need for gradual and individualized training resumption. ”

Recent findings suggest that after a long period of inactivity such as the COVID-19 lockdown, athletes require a carefully structured retraining program lasting at least 6–8 weeks to restore their physiological and performance capacities [3]. Independent of the cause of training cessation, previous research has shown that the longer the inactivity period, the greater the loss in endurance, strength, and neuromuscular coordination, thereby emphasizing the importance of carefully structured retraining strategies for athletes returning after prolonged inactivity [4]. Sudden increases in training volume and intensity after a period of confinement may lead to an overload of physiological systems, raising the risk of injuries. Properly managed progressive loading, baseline physical assessments, and monitoring tools such as heart rate variability and perceived exertion scales are essential to safely reinstate competitive training [5, 6].

Suggestion: Although there is no similar research on this subject during the Covid-19 period, a comparison should be made in the discussion section with studies that express how many weeks and which methods are applied to athletes, especially after injuries or when the athlete is away from the field for a long time.

Response: We discussed and compared findings with studies focused on recovery strategies after injuries and long-term detraining, expanding the context beyond COVID-19.

Grazioli et al. (2020) reported that an 8-week quarantine period negatively impacted football players' physical performance, including power, sprinting, body mass, and body fat mass, and suggested that athletes needed longer recovery periods than regular training programs. Recent findings suggest that after a long period of inactivity such as the COVID-19 lockdown, athletes require a carefully structured retraining program lasting at least 6–8 weeks to restore their physiological and performance capacities (Valenzuela et al., 2021).

Suggestion: Although the relevant research was conducted in a specific process such as Covid-19, the findings obtained are very important for athletes who have to stop training for a long time. Therefore, more space should be given to similar studies suitable for the specified athlete profile.

Response: We included new paragraphs connecting our findings to broader cases (e.g., injury-related absences), discussing the applicability of the results.

“Several studies [32-34] have shown that stress and anxiety affect the autonomic nervous system in competition period, resulting in a decline in heart rate variability values.”

“It was reported that short-term detraining induces significant increases in body fat mass (+21%) and reductions in muscle mass (~4%) among elite athletes, which may negatively affect performance and agility [7].”

Recent findings suggest that after a long period of inactivity such as the COVID-19 lockdown, athletes require a carefully structured retraining program lasting at least 6–8 weeks to restore their physiological and performance capacities (Valenzuela et al., 2021). Independent of the cause of training cessation, previous research has shown that the longer the inactivity period, the greater the loss in endurance, strength, and neuromuscular coordination, thereby emphasizing the importance of carefully structured retraining strategies for athletes returning after prolonged inactivity (Mujika & Padilla, 2000).

Suggestion: In the relevant studies, the importance of the parameters examined for the wrestling branch and how these parameters can affect performance with rapid recovery should be included in the foresight and similar studies. -More suggestions should be made regarding the results of the research, specific to the wrestling branch.

Response: In response, we have expanded the discussion section by emphasizing how the key physiological and performance parameters assessed in our study, such as anaerobic endurance, maximal strength, and dynamic balance, are critical for rapid recovery and optimal performance in wrestling. We have also incorporated references to support the wrestling-specific importance of these parameters. Accordingly, we have added practical recommendations based on our findings, specifically tailored to the needs of wrestlers. These include implementing periodized retraining programs emphasizing anaerobic conditioning, technical drills under fatigue, and injury prevention strategies during return-to-training phases after long-term inactivity.

“Anaerobic endurance, maximal strength, and dynamic balance are fundamental components for wrestling performance. Rapid recovery of these parameters following a period of detraining is essential for maintaining explosive movement patterns and technical efficiency during competition.”

“Sudden increases in training volume and intensity after a period of confinement may lead to an overload of physiological systems, raising the risk of injuries. Properly managed progressive loading, baseline physical assessments, and monitoring tools such as heart rate variability and perceived exertion scales are essential to safely reinstate competitive training (Casais-Martinez et al., 2020; Impellizzeri et al., 2020). “

“The fact that the lactate level is lower despite the increase in the throws performed in the SWFT test indicates that the lactate threshold and lactate removal capacity were also developed.”

Reviewer #2:

We sincerely thank the reviewer for their valuable time, insightful suggestions, and constructive feedback, all of which have significantly contributed to improving the quality and clarity of our manuscript. We carefully considered each comment and made the necessary revisions accordingly. Below, we provide a detailed, point-by-point response. The authors’ responses are written in italics. Where appropriate, changes made in the manuscript have been highlighted in green.

Suggestion: Page 12, line 25 – so you have mentioned that HRV is the key variable observed and then you don’t mention the effects of detraining and retraining on it?

Response: We appreciate the reviewer’s insightful comment. We have revised the abstract to explicitly state that heart rate variability (HRV) decreased during the training cessation period, progressively improved during the retraining phase, and subsequently declined again during the competition phase, approaching values observed during the detraining period. This highlights the sensitivity of HRV to training load fluctuations and competitive stress. In addition to revising the main body of the abstract, we have also updated the conclusion section to emphasize that HRV was negatively impacted not only during detraining but also during the competition phase. This addition highlights the practical importance of HRV monitoring for managing training loads and optimizing athlete recovery and readiness for competition.

“Heart rate variability decreased during the detraining period, progressively improved throughout the 8-week retraining, and subsequently declined during the competition phase, reaching levels similar to those observed during training cessation.”

“Moreover, HRV monitoring revealed that autonomic nervous system balance was compromised during both the detraining and competition phases, underlining the need for careful training load management to optimize recovery and performance readiness.”

Suggestion: Line 14, line 62 – you mean the prevention of social gatherings?

Response: In the revised manuscript, we have specified that the COVID-19 lockdown measures involved the prevention of social gatherings, result

---

## [Decision Letter · Decision Letter 1]

Dear Dr. Melekoglu,

Thank you for submitting your manuscript to PLOS ONE. After careful consideration, we feel that it has merit but does not fully meet PLOS ONE’s publication criteria as it currently stands. Therefore, we invite you to submit a revised version of the manuscript that addresses the points raised during the review process.

We look forward to receiving your revised manuscript.

Kind regards,

Ozkan Isik

Academic Editor

PLOS ONE

Journal Requirements:

Additional Editor Comments:

Reviewer 1 has accepted your article. However, Reviewer 2 wants you to make the following minor changes. Please make the requested corrections one by one and send the 2nd round version of your article to the reviewer along with your responses.

Page 18, line 26 – authors failed to report how long the detraining phase was in the abstract section. If you state that retraining period was 8 weeks, it would be of great value to also know how long was the detraining phase.

Page 19, line 44 – “During these periods of partial or complete training interruption, it has been suggested that the organism’s responsiveness to exercise is significantly prolonged”. I am not sure what this sentence means. Refine for clarity.

Page 19, line 46 – “Athletes facing training interruptions may encounter risk factors that negatively impact performance, including declines in muscle strength, muscle size, cardiovascular endurance, as well as impairments in neuromuscular coordination and flexibility.” Provide citation(s).

Page 21, line 79 – “Proper training methods can minimize the risk of injury, prevent long-term losses in muscle size and strength, and facilitate a faster and more reliable return to optimal fitness and performance levels. Structured and individualized retraining programs are crucial for minimizing injury risk and accelerating performance recovery in elite 75 wrestlers following prolonged periods of training cessation. Given the detrimental effects of training cessation on athletic performance, it is essential for athletes to implement appropriate strategies to mitigate performance declines.” Move this section to discussion. It deviates from the key points in the Introduction and it makes too long.

Page 24, line 148 – which device did you use to measure HRV?

Page 24, line 157 – Turkish national team I’d assume? Mention this. It’s important.

Page 26, line 199 – they COULD not… Stay in the past tense.

Page 26, line 213 – WERE required. The same principle applies.

Page 28, line 239 – “The participant was performed three sets”. As I said in the initial review, this manuscript would benefit greatly if the native English-speaking person reads it.

Best regards,

Reviewers' comments:

Reviewer's Responses to Questions

**Comments to the Author**

Reviewer #1: All comments have been addressed

Reviewer #2: All comments have been addressed

2. Is the manuscript technically sound, and do the data support the conclusions?

Reviewer #1: Yes

Reviewer #2: Yes

3. Has the statistical analysis been performed appropriately and rigorously?

Reviewer #1: Yes

Reviewer #2: I Don't Know

4. Have the authors made all data underlying the findings in their manuscript fully available?

Reviewer #1: Yes

Reviewer #2: Yes

5. Is the manuscript presented in an intelligible fashion and written in standard English?

Reviewer #1: Yes

Reviewer #2: Yes

Reviewer #1: The arrangements I have previously stated have been fulfilled. It is believed that the article will be an important research in this field. The article is suitable for publication.

Reviewer #2: Page 18, line 26 – authors failed to report how long the detraining phase was in the abstract section. If you state that retraining period was 8 weeks, it would be of great value to also know how long was the detraining phase.

Page 19, line 44 – “During these periods of partial or complete training interruption, it has been suggested that the organism’s responsiveness to exercise is significantly prolonged”. I am not sure what this sentence means. Refine for clarity.

Page 19, line 46 – “Athletes facing training interruptions may encounter risk factors that negatively impact performance, including declines in muscle strength, muscle size, cardiovascular endurance, as well as impairments in neuromuscular coordination and flexibility.” Provide citation(s).

Page 21, line 79 – “Proper training methods can minimize the risk of injury, prevent long-term losses in muscle size and strength, and facilitate a faster and more reliable return to optimal fitness and performance levels. Structured and individualized retraining programs are crucial for minimizing injury risk and accelerating performance recovery in elite 75 wrestlers following prolonged periods of training cessation. Given the detrimental effects of training cessation on athletic performance, it is essential for athletes to implement appropriate strategies to mitigate performance declines.” Move this section to discussion. It deviates from the key points in the Introduction and it makes too long.

Page 24, line 148 – which device did you use to measure HRV?

Page 24, line 157 – Turkish national team I’d assume? Mention this. It’s important.

Page 26, line 199 – they COULD not… Stay in the past tense.

Page 26, line 213 – WERE required. The same principle applies.

Page 28, line 239 – “The participant was performed three sets”. As I said in the initial review, this manuscript would benefit greatly if the native English-speaking person reads it.

**Do you want your identity to be public for this peer review?** For information about this choice, including consent withdrawal, please see our Privacy Policy

Reviewer #1: No

Reviewer #2: No

---

## [Author Response · Author response to Decision Letter 2]

28 May 2025

Reviewer #2:

We sincerely thank you for your valuable comments and suggestions. We have carefully addressed each of your points and revised the manuscript accordingly. Below, we provide a detailed, point-by-point response. The authors’ responses are written in italics. Where appropriate, changes made in the manuscript have been highlighted in yellow.

Suggestion: Page 18, line 26 – authors failed to report how long the detraining phase was in the abstract section. If you state that retraining period was 8 weeks, it would be of great value to also know how long was the detraining phase.

Response: Thank you for your valuable comment. As you rightly pointed out, clearly stating the duration of the detraining phase adds clarity and value to the abstract and overall interpretation. Although the 14-week detraining period was already mentioned in the background section in abstract, we have now revised the conclusion section to explicitly include the detraining duration in the relevant sentence for improved clarity. The revised sentence now reads:

“Moreover, HRV monitoring revealed that autonomic nervous system balance was compromised during both the 14-week detraining and the subsequent competition phases, underlining the need for careful training load management to optimize recovery and performance readiness.”

We hope this revision addresses your concern.

Suggestion: Page 19, line 44 – “During these periods of partial or complete training interruption, it has been suggested that the organism’s responsiveness to exercise is significantly prolonged”. I am not sure what this sentence means. Refine for clarity.

Response: Thank you for your comment. We agree that the original sentence lacked clarity and could be misinterpreted. To improve the precision of the statement, we have revised the sentence on Page 19, line 44 as follows:

“During periods of partial or complete training interruption, the body’s physiological responsiveness to exercise stimuli may become diminished or delayed, leading to a slower return to previous performance levels upon retraining.”

We believe this updated version more accurately reflects the intended meaning and appreciate your helpful suggestion.

Suggestion: Page 19, line 46 – “Athletes facing training interruptions may encounter risk factors that negatively impact performance, including declines in muscle strength, muscle size, cardiovascular endurance, as well as impairments in neuromuscular coordination and flexibility.” Provide citation(s).

Response: We have added relevant citations to the sentence on Page 19, line 46, which now reads:

“Athletes facing training interruptions may encounter risk factors that negatively impact performance, including declines in muscle strength, muscle size, cardiovascular endurance, as well as impairments in neuromuscular coordination and flexibility (Alvurdu et al., 2022; Bosquet et al., 2013; Mujika & Padilla, 2000b).

Suggestion: Page 21, line 79 – “Proper training methods can minimize the risk of injury, prevent long-term losses in muscle size and strength, and facilitate a faster and more reliable return to optimal fitness and performance levels. Structured and individualized retraining programs are crucial for minimizing injury risk and accelerating performance recovery in elite 75 wrestlers following prolonged periods of training cessation. Given the detrimental effects of training cessation on athletic performance, it is essential for athletes to implement appropriate strategies to mitigate performance declines.” Move this section to discussion. It deviates from the key points in the Introduction and it makes too long.

Response: As recommended, we have moved the relevant section from the Introduction to the Discussion to maintain a clearer focus on the key points and improve the overall structure and flow of the manuscript. We believe this adjustment enhances both the coherence and readability of the paper.

Suggestion: Page 24, line 148 – which device did you use to measure HRV?

Response: We have revised the sentence to include the specific device used for HRV measurement. The updated sentence now reads:

“HRV data were collected using a Polar H10 heart rate chest strap sensor.”

Suggestion: Page 24, line 157 – Turkish national team I’d assume? Mention this. It’s important.

Response: We have revised the sentence to clearly indicate that the participants were members of the Turkish National Wrestling Team. The updated sentence now reads:

“This study included 20 male volunteer wrestlers (aged 21.75 ± 3.11 years) who were members of the Turkish National Wrestling Team and had been training for at least 10 years.”

Suggestion: Page 26, line 199 – they COULD not… Stay in the past tense.

Response: As suggested, we have revised the sentence to remain consistent with the past tense. The corrected version now reads:

“Due to the constraints imposed by the pandemic, the wrestlers unable to perform a structured training program during the long-term training cessation period.”

Suggestion: Page 26, line 213 – WERE required. The same principle applies.

Response: As suggested, we have revised the sentence to remain consistent with the past tense. The corrected version now reads:

“Wrestlers were required to attend at least 90 % of the training sessions. The weekly breakdown of the 8-week training program is provided in Table 1.”

Suggestion: Page 28, line 239 – “The participant was performed three sets”.

Response: As suggested, we have revised the sentence to remain consistent with the past tense. The corrected version now reads:

“Each participant performed three sets of 30-second dummy throws, separated by 20-second rest intervals.”

Suggestion: As I said in the initial review, this manuscript would benefit greatly if the native English-speaking person reads it.

Response: As per your suggestion, the manuscript has been thoroughly reviewed and revised by a native English speaker to improve its linguistic quality, clarity, and fluency.”

---

## [Editor Report · Decision Letter 2]

The impact of 8-week re-training following a 14-week period of training cessation on Greco-Roman Wrestlers

PONE-D-25-10641R2

Dear Dr. Melekoglu,

We’re pleased to inform you that your manuscript has been judged scientifically suitable for publication and will be formally accepted for publication once it meets all outstanding technical requirements.

Kind regards,

Ozkan Isik

Academic Editor

PLOS ONE
---

## [Editor Report · Acceptance letter]

PONE-D-25-10641R2

PLOS ONE

Dear Dr. Melekoglu,

I'm pleased to inform you that your manuscript has been deemed suitable for publication in PLOS ONE. Congratulations! Your manuscript is now being handed over to our production team.

Kind regards,

on behalf of

Professor Ozkan Isik

Academic Editor

PLOS ONE